# The Potential of Insects as Alternative Sources of Chitin: An Overview on the Chemical Method of Extraction from Various Sources

**DOI:** 10.3390/ijms21144978

**Published:** 2020-07-15

**Authors:** Nurul Alyani Zainol Abidin, Faridah Kormin, Nurul Akhma Zainol Abidin, Nor Aini Fatihah Mohamed Anuar, Mohd Fadzelly Abu Bakar

**Affiliations:** 1Faculty of Applied Sciences and Technology, Universiti Tun Hussein Onn Malaysia, Johor 86400, Malaysia; alyanizainolab@gmail.com (N.A.Z.A.); akhma1993@gmail.com (N.A.Z.A.); fsolehah940301@gmail.com (N.A.F.M.A.); fadzelly@uthm.edu.my (M.F.A.B.); 2Centre of Research on Sustainable Uses of Natural Resources, Universiti Tun Hussein Onn Malaysia, Johor 86400, Malaysia

**Keywords:** chitin, chitosan, chemical extraction, deproteinization, demineralization

## Abstract

Chitin, being the second most abundant biopolymer after cellulose, has been gaining popularity since its initial discovery by Braconot in 1811. However, fundamental knowledge and literature on chitin and its derivatives from insects are difficult to obtain. The most common and sought-after sources of chitin are shellfish (especially crustaceans) and other aquatic invertebrates. The amount of shellfish available is obviously restricted by the amount of food waste that is allowed; hence, it is a limited resource. Therefore, insects are the best choices since, out of 1.3 million species in the world, 900,000 are insects, making them the most abundant species in the world. In this review, a total of 82 samples from shellfish—crustaceans and mollusks (n = 46), insects (n = 23), and others (n = 13)—have been collected and studied for their chemical extraction of chitin and its derivatives. The aim of this paper is to review the extraction method of chitin and chitosan for a comparison of the optimal demineralization and deproteinization processes, with a consideration of insects as alternative sources of chitin. The methods employed in this review are based on comprehensive bibliographic research. Based on previous data, the chitin and chitosan contents of insects in past studies favorably compare and compete with those of commercial chitin and chitosan—for example, 45% in *Bombyx eri*, 36.6% in *Periostracum cicadae* (cicada sloughs), and 26.2% in *Chyrysomya megacephala*. Therefore, according to the data reported by previous researchers, demonstrating comparable yield values to those of crustacean chitin and the great interest in insects as alternative sources, efforts towards comprehensive knowledge in this field are relevant.

## 1. Introduction

Chitin and its derivatives represent a well-reviewed biopolymer with many beneficial applications. The preparations for chitin and its derivatives as a biomaterial vary according to process conditions and potential applications. However, their main sources are crustaceans, and research on alternative sources is still developing. In this article, we comprehensively review the preparation of chitin and its derivatives, especially chitosan, via chemical extraction, for a comparison of the optimal demineralization and deproteinization processes with a consideration of insects as alternative sources. Chitin is the basic structure and a major constituent of the cell wall of many fungi, insect exoskeletons, and crustacean shells. It largely exists in waste from the processing of marine food products, such as shrimp, prawn, crab, lobster, crayfish, squid, cuttlefish, and barnacles [1]. Chitin extraction has been well-known since its first isolation in 1811 by Henri Braconnot from some of the higher fungi, and chitin is the earliest known polysaccharide [2]. On the other hand, Albert Hoffman was the first researcher to determine the structure of chitin [3]. His main interest was the chemistry of plants and animals, and he later conducted important research during his study on the chemical structure of common animal substances and hence the discovery of chitin, for which he received his doctorate, with distinction, in the spring of 1929 [4].

Chitin is a biopolymer and is the most abundant biopolymer after cellulose, with a production of approximately 10^10^–10^12^ tons annually [5,6,7]. Chitin has the same chemical structure as cellulose— a plant fiber [2]. It is a linear polysaccharide composed of β-1,4 linked with N-acetylglucosamine (GlcNAc) units (to be precise, 2-(acetylamino)-2-deoxy-D-glucose) and occurs naturally in three polymorphic forms, with different orientations of the microfibrils, known as α, β, and γ chitin [5,6].

Chitin isolates from crustaceans, other aquatic invertebrates, and arthropods, especially those with a hard exterior, usually come in the α-form. This is because the chains are aligned in an anti-parallel structure, thus making the structure more stable owing to stronger and rigid hydrogen bonding. Meanwhile, β-form chains are arranged in a parallel fashion, and most chitin sources with this β-form are obtained from mollusks, such as squid pens [8]. The γ-form contains two parallel strands and one anti-parallel strand of chitin and is found in cocoons of insects. Conversion from the β-form to the α-form is possible, but not the reverse. It has been reported that the isolation of chitin from different sources is affected by the source and the percentage of chitin present in the source, and it was found that the crystallinity, purity, and polymer chain arrangement vary, according to the source. Crab, shrimp, and crayfish have been preferred for the commercial production of chitin, but new alternative chitin sources, such as fungus and insects, could be exploited [9,10,11].

Chitosan is derived from chitin by removing a sufficient number of acetyl groups (CH_3_–CO) for the molecule to be soluble in most diluted acids. Chitosan is a fiber-like substance and a homopolymer of β-(1→4)-linked N-acetyl-D-glucosamine. The actual difference between chitin and chitosan is the acetyl content of the polymer. Chitosan, which has a free amino group, is the most useful derivative of chitin [12]. Chitosan is a modified natural carbohydrate polymer that has been found in a wide range of natural sources, such as crustaceans, mollusks, fungi, insects, and some algae [13]. Chitosan is a non-toxic, biodegradable polymer of a high molecular weight and is very similar to cellulose in terms of its chemical structure. The only difference between chitosan and cellulose is the amine (-NH_2_) group in the C-2 position of chitosan instead of the hydroxyl (-OH) group found in cellulose. However, unlike plant fiber, chitosan possesses positive ionic charges that give it the ability to chemically bind with negatively charged fats, lipids, cholesterol, metal ions, proteins, and macromolecules. In this respect, chitin and chitosan have attained increasing commercial interest as suitable materials due to their excellent properties, including their biocompatibility, biodegradability, adsorption, and abilities to form films and chelate metal ions [14,15,16].

Figure 1a indicates the chemical configuration of chitin. Figure 1b shows the structure of the chitin molecule, and presents two of the N-acetylglucosamine units that repeat to form long chains in the β-(1→4)-linkage. These units form covalent β-(1→4)-linkages (similar to the linkages between glucose units forming cellulose). Therefore, chitin may be described as cellulose with one hydroxyl group on each monomer replaced with an acetyl amine group. This allows for increased hydrogen bonding between adjacent polymers, giving the chitin–polymer matrix an increased strength [1,12,17].

In its pure, unmodified form, chitin is translucent, pliable, resilient, and quite tough. In most arthropods, however, it is often modified, occurring largely as a component of composite materials, such as in sclerotin, a tanned proteinaceous matrix, which forms much of the exoskeleton of insects. Combined with calcium carbonate (CaCO_3_), as in the shells of crustaceans and mollusks, chitin produces a much stronger composite. This composite material is much harder and stiffer than pure chitin and is tougher and less brittle than pure CaCO_3_. Another difference between pure and composite forms can be seen by comparing the flexible body wall of a caterpillar (mainly chitin) to the stiff, light elytron of a beetle (containing a large proportion of sclerotin) [14,15,16,18].

In addition, chitin and chitosan have the ability to bond with different links, forming materials such as fibers, hydrogels, beads, sponges, and membranes. Chitosan has been used in several fields, such as agriculture, food protection [19], biomedicine [20], cosmeceuticals [21], and pharmaceuticals [22] as drug delivery systems or in drug formulations [15,16,18,23,24,25,26,27]. Sources of chitin and chitosan that are used in such beneficial applications mostly consist of shellfish, mainly crustaceans [28,29]. However, interest in chitin and chitosan from insects as alternative sources other than crustaceans has increased steadily over the last several decades. A handful of researchers have paid attention to and elucidated the process of extraction, optimization, and yield, as well as the characterization of chitin and chitosan from various sources, especially insects. Hence, insects are gaining popularity as more researchers study them. They represent the best choice as an alternative since out of 1.3 million species in the world, 900,000 are insects, making them the most abundant species in the world [30,31]. Therefore, the aim of this paper was to review the extraction method of chitin and chitosan for a comparison of the optimal demineralization and deproteinization processes, with a consideration of insects as alternative sources of chitin.

Even though chitin and chitosan have been called our “last biomass resource” and are expected to lead to a new functional polymer, their utilization is scarce, and they have hardly been explored. Though a variety of interesting biological activities have been reported throughout the years, practical application has lagged. One of the main reasons for this is that these biological activities are not specific to chitosan; such activities are also found in other materials. The second reason is the issue of cost, since chitosan is relatively expensive (20–30 US dollars per kg). If a specific biological activity was found to be unique to chitosan materials, practical utilization would be encouraged despite the cost, especially for biomedical use. Chitin and chitosan are structurally similar to heparin, chondroitin sulfate, and hyaluronic acid, which are all biologically important mucopolysaccharides in all mammals. These mucopolysaccharides are anionic polymers owing to substituent carboxyl and sulfuryl groups. On the other hand, chitosan is almost the only cationic polysaccharide in nature, and it is nontoxic and biodegradable in the human body [6]. This special property is worth noting in regard to biomedical applications [32]. However, since chitosan does not dissolve in neutral and basic aqueous media, its biomedical use is limited. The chemical modification of chitosan provides derivatives that are soluble at a neutral and basic pH. Moreover, chemical modification can be used to attach various functional groups and to control hydrophobic, cationic, and anionic properties [33,34]. Further studies and the development of chitin, chitosan, and their derivatives for use in applied biomaterials need to be carried out [5]. These points can be considered in future developments in the field of biomaterials from insects.

## 2. Result and Discussion

Chitin is gaining popularity, especially because many beneficial traits are being discovered for fertilizers [35,36,37], food additives [38], emulsifying agents [39,40], and surgical [41,42,43] and medicinal [44] applications, as well as in agricultural [45], pharmaceutical [46,47], and even cosmeceutical [21,48,49,50] fields. The most common extraction methods are biological [51,52,53,54] and chemical [10,17] treatments. As mentioned above, chemical treatment involves two major steps with an optional treatment. Demineralization is an acidic step that removes minerals associated with the basic structure of the exoskeleton, and deproteinization is a basic step that removes the proteins bound together with all constituents. Generally, for comparison, the parameters for acidic treatment (concentration, temperature, time, and solution-to-solid ratio) used for chemical extraction from insects are moderate compared to crustaceans’ chitin isolation requirements. This is because insects have lower levels of inorganic material (less than 10%) compared to crustacean shells (20%–40%) [13].

There are numerous alternative methods of extraction for chitin, where their sole purpose is to remove impurities and foreign organic matter, including protein and minerals, because it is naturally formed in the structure of the exoskeleton [10,55]. However, detrimental effects on the molecular weight and the degree of acetylation are unavoidable with any of the extraction processes.

For demineralization, also known as decalcification, hydrochloric acid (HCl) is the most commonly used acid by far. Although it may be the cause of the detrimental effects on the intrinsic properties of the purified chitin, it remains the most used decalcifying agent for both the laboratory and industrial-scale production of chitin. It is widely used at various concentrations. Hence, chemical extraction is the most convenient and effective method when it is optimized, as optimization of the extraction can minimize the degradation of chitin and reduce the impurity levels to a satisfactory level for specific applications [56,57]. The demineralization process is conducted via inorganic or organic acids, such as hydrochloric acid (HCl), nitric acid (HNO_3_), sulfuric acid (H_2_SO_3_), acetic acid (CH_3_COOH), and formic acid (HCOOH), with HCl being the most commonly used reagent at a concentration of 0.25–4 M at 21,100 °C, from 15 min to 48 h. The solid-to-solvent ratio is employed in the range of 1:9–1:50 (w/v). Although other effective reagents have been reported, alkaline treatment for deproteinization is a commonly used method of protein extraction. [1,11,12]. Another trend involving the use of enzymes is emerging, representing a newly found method for protein treatment [58,59,60]. However, the residual protein is higher, with a reaction time similar to or longer than that of the chemical method [61]. These drawbacks make the enzymatic degradation of protein method less likely to be applied [54,62]. In addition, it is more expensive compared to chemical treatment [63]. For protein treatment, chemicals such as potassium hydroxide (KOH), sodium carbonate (Na_2_CO_3_), potassium carbonate (K_2_CO_3_), calcium hydroxide (CaOH_2_), and sodium sulfate (Na_2_SO_4_), at various concentrations in aqueous solutions, are applied, with NaOH as a commonly used base. The alkaline concentration for deproteinization can be seen to range from 0.025 to 4 M at 25–150 °C, from 20 min to 96 h, where the solid-to-solvent ratio ranges from 1:5 to 1:100 (*w*/*v*). In addition, a new trend has recently emerged for assisting with the process of extraction. Aside from chemical and biological treatments, physical treatment has also gained popularity. The basic treatment still involves acids and bases as the main media, but is assisted by ultrafast microwave [64] treatment and ultrasound treatment [65].

Decolorization is performed using several methods: alcohol washing, chloroform and alcohol solutions, phosphorus pentoxide solutions, potassium permanganate, a mixture of oxalic acid and sulfuric acid solutions, and sodium hypochlorite solutions. Decolorization or color bleaching is an optional step, though almost no researcher skips this step due to certain factors. Exoskeletons or crustaceans naturally have pigments that give each characteristic a different color. Decolorization can occur, or the pigment can be destroyed, using oxidants such as potassium permanganate (KMnO_4_), with or without hydrogen peroxide (H_2_O_2_), sodium hypochlorite (NaClO), phosphorus pentoxide (P_2_O_5_), sulfur dioxide (SO_2_), and sodium carbonate (Na_2_CO_3_). In addition, solvents are often used in pigment removal. Examples of solvents used include acetone, chloroform, ethyl acetate, and ethanol. However, when pigment recovery is important for economical accounts, for instance, carotenoids, solvent extraction is used to recover pigments [66]. In other words, it is not necessarily important to completely remove pigments, and there are cases where pigments are economically important [66].

Deacetylation is an extension procedure; by removing the acetyl group from the chitin structure, chitosan can be derived. From a chemical point of view, either acids or alkalis can be used to deacetylate chitin. However, glycosidic bonds are very susceptible to acid. Therefore, alkali deacetylation is used more frequently [67,68]. Deacetylation is investigated using seven factors: the alkali reagent, along with its concentration, its temperature, and its reaction time; the use of successive baths; atmospheric conditions; and the use of sodium borohydride, a reducing agent [69].

Over the past few decades, the extraction of chitin has evolved according to its application, and continuous improvements have been made to increase the efficiency of the extraction method, as well as to match its application. Figure 2 is a flow chart of the overall extraction method for chitin summarized from Table 1, Table 2 and Table 3, starting from an input powdered sample to an output crude chitin, followed by the deacetylation of chitosan.

Generally, the higher the concentration of alkali for deproteinization is (including parameters such as the pH, solid-to-liquid ratio, time, and temperature), the lower the resulting molecular weight of chitin ought to be. The same conditions are applied to decalcification via acid treatment [7]. Therefore, for an acid-sensitive material, physical property modifications are likely possible due to degradation via several pathways, such as hydrolytic depolymerization and deacetylation, including heat degradation, when introduced to highly concentrated acid. Therefore, an optimized condition will help extract the highest possible yield [7,57,70].

### 2.1. Commercial Preparation of Chitin

In 1970, Peniston and Johnson patented a method for treating an aqueous medium with chitosan and other derivatives of chitin to remove impurities [71]. Taking the next step, in 1975, they patented another method of recovering chitosan and other by-products from shellfish waste [72], in addition to a process for the depolymerization of chitosan [73]. This invention allowed for a treatment for removing protein, whereby a strong base is used for purposes such as reducing the viscosity, increasing the solubility, and generally changing poly-electrolyte characteristics. A commercial demineralization process of preparing chitin from crustacean shells was patented by them in 1978. In order to produce chitin with a higher purity, another step was included in the purification process. The additional process concerned the removal of minerals, known as demineralization (removing CaCO_3_), and involved the use of a strong acid [74]. In 1980, they invented a novel process of manufacturing chitosan from chitin with reduced temperatures, increased reaction rates involving higher alkali-to-solid deacetylation ratios, and a quiescent air-expelled final deacetylation step [75]. For the last few years, chitin and its derivatives have been commercially applied to biomedicine [76,77,78,79,80], pharmaceuticals [22], and even weight-loss agents [81,82]. The purification of commercial chitin usually involves nanotechnology. Chitin and chitosan nanoparticles usually act as a carrier or co-carrier and will associate with other biopolymers and bioactive compounds [83].

### 2.2. Chemical Extraction of Chitin and Chitosan from Insects as an Alternative to Commercial Chitin from Crustaceans

In Table 1, Table 2 and Table 3, a comprehensive bibliography of chemical extraction methods for chitin and chitosan is given. A total of 82 samples were collected from crustaceans, mollusks, and other aquatic invertebrates (n = 46), insects (n = 23), and others (n = 13). They were collected and studied for their chemical treatment for extracting chitin and its derivatives. Different samples were selectively collected and reviewed (insects, arachnids, crustaceans, and others). As its economical usage grows exponentially, research on underutilized materials and improvised methods continues to flourish [66]. Crustaceans were originally considered as the main sources of chitin. However, with the increasing demand, the supply needed to increase. Alternative sources of chitin and its derivatives are emerging and continuing to grow alongside the rise of new areas in the utilization and application of this most important biopolymer, second only to cellulose. Hence, new and underutilized materials are being explored as alternative sources. Insects show great potential as an alternative [66].

Zhang et al. isolated chitin from beetle larva cuticles and silkworm pupa exuviae using chemical treatment with 1 N HCl at 100 °C for 20 min for the removal of inorganic minerals, mostly catechols. For removing protein, they used 1 N NaOH at 80 °C for 36 or 24 h and refluxed it with 0.4% Na_2_CO_3_ for 20 h. The residue was washed with distilled water and dried on P_2_O_5_ in a vacuum. The result was a 15%–20% yield of chitin [10]. Chitosan extraction was performed by treatment in 40% NaOH containing sodium borohydrite (NaBH_4_) [10]. They found that insect chitin, compared to shrimp chitin, was more vulnerable and more easily broken down when treated with 6 N HCl and the enzyme chitinase. After treatment with 2 N HCl at 100 °C, the insect chitin’s crystallinity increased. The deacetylation of chitin from insects was easier than that from crustaceans. This was shown when about 94% of the N-acetyl groups were removed after only one treatment with 40% NaOH for 4 h at 110 °C. The deproteinized sample was treated with 2 N HCl, and 55% of the N-acetyl groups of silkworm chitin were removed under the same conditions. As a result, it was reported that beetle chitin showed a higher affinity for chitinase than shrimp chitin [10].

Silkworm yielded an average value of 15%–20%. The isolated treatment involved demineralization (HCl, 1 N, 100 °C for 20 min) and deproteinization steps (NaOH, 1 N, 80 °C for a period of 1–2 days, in addition to a step involving reflux with 0.4% NaCO3 for 20 h) [10]. In reference to Zhang et al., Paulino’s team reported that the chitin and chitosan they isolated gave yield values of 2.59%–4.23% [84]. Less chitin was yielded from silkworm, but it produced high-purity and porous chitosan [84]. This report can be supported by Huet et al., who purified chitin from *B. eri* and yielded 45% of high-purity chitin with only one extraction step [85]. This high chitin yield is also due to the low mineral content of insect larvae, supporting the use of insects as alternative sources of biopolymer materials. Therefore, insects have a competitive potential as an alternative source of chitin and its derivatives.

Matjan et al. described the isolation process of chitin, as well as its characterization, from bumblebees (*Bombus terrestris*); the process arose from the biotechnological production of bumblebees used for vegetable and fruit pollination [55]. In addition, they compared the physicochemical properties of insect chitin (bumblebee) with crustacean chitin, which was commercially purchased *Pandalus borealus* shells (shrimp). In their report, insect chitin was prepared using dead bumblebees treated with 1 M HCl and 1 M NaOH. The bumblebee chitin was compared with the shrimp chitin. The comparison was based on their characteristics studied via elemental analysis, Fourier-transform infrared (FTIR) spectroscopy, solid-state 13C cross-polarization magic-angle-spinning nuclear magnetic resonance (^13^CP/MAS-NMR) spectroscopy, and confocal microscopy. It was found that the chitin yield from the bumblebee was lower than that from shrimp. On the other hand, bumblebee chitin and chitosan exhibited a finer texture. Chitin extracted from both sources (bumblebee and shrimp) showed identical spectra. However, since solid-state ^13^C NMR spectroscopy was sensitive to changes in the local structure, the result for bumblebee chitin had a 5% lower degree of acetylation and was characterized by a fine membrane texture. In conclusion, the extraction method described allowed the end-product to be isolated with a high chemical purity [55].

In a study conducted by Sajomsang and Gonil, cicada slough was proposed as an alternative source of chitin [86]. Cicada slough chitin was isolated chemically; the demineralization process was conducted via 1 M HCl and deproteinization was conducted via 1 M NaOH [86]. The depigmentation of brown chitin from cicada sloughs was performed using 6% sodium hypochlorite as an oxidizing agent. It was reported that the insect cicada chitin yielded a higher result in terms of the recovery percentage, when compared with chitin from rice-field crab shells (36.6% vs. 15.2%, respectively) [86]. This was because of the low level of inorganic materials naturally occurring in insect chitin. For the determination of the degree of acetylation, elemental analysis (EA), 1H NMR, and ^13^CP/MAS-NMR were used. However, cicada slough chitin was found to have a lower degree of acetylation, crystallinity, and thermal properties. Sajomsang and Gonil agreed that cicada slough exhibited potential as an alternative source, not only for these reasons, but also for its ecological safety [86].

Another alternative source of chitin was proposed by Liu et al.: adult *Holotrichia parallela*, the motschulsky beetle [11]. The yield of chitin from adult *H. parallela* was shown to be around 15%. The yields of chitin from other insects varied with species and their development stages. The characteristics of chitin from adult *H. parallela* were reported to be similar to those of commercial chitin from shrimp by infrared (IR), X-ray diffraction (XRD), scanning electron microscope (SEM), and elemental analysis. The chitin extracted from adult *H. parallela* is thus suitable for chitosan production. The large numbers of *H. parallela* adults captured for the control of this pest in fields every year provide an abundant source of chitin. In addition, it was concluded that attempts to domesticate the beetle for a stable supply have a high potential for relieving the impact on ecological systems in the near future [11].

In a report by Song et al., blowfly larvae were isolated and studied as an alternative source of chitosan. The physical properties of blowfly larvae chitosan were evaluated via approaches similar to those of previous researchers, with methods including color-change identification, molecular weight determination, elemental analysis, FTIR, ^13^CP/MAS-NMR, and SEM [87]. This was the first report on the isolation of chitosan from blowfly larvae. The results obtained show that chitosan from blowfly has a lower molecular weight and a higher degree of deacetylation (DDA), and they also studied the antioxidant properties, which were superior compared to those of commercial chitosan [87]. The results demonstrated a promising alternative for chitosan. Aside from having a lower molecular weight, a higher DDA, and superior antioxidant properties, blowfly is non-seasonal, so there is no seasonal limit for a continuous supply [87].

Kaya et al. studied both adult and larval potato beetle (*Leptinotarsa decemlineata*) via the same chemical extraction process. Both adults and larvae were compared for chitin isolation, and the results showed that the structure of chitin for both was in an alpha form, as in other insects [11,55,86,88]. Thermogravimetric analysis (TGA) revealed similar traits for both. Meanwhile, the maximal decomposition rate (DTGmax) values of chitin were different. This study determined that the chitin from adult potato beetles exhibits a higher thermal stability compared to larval chitin. It was reported that the dry weight of the chitin isolated from adults was relatively higher than that from larval chitin. The same results were obtained for chitosan extraction: adult chitin exhibited a higher yield. Elemental analysis results showed that the DDA value of adult chitin was closer to 100% compared to larval chitin. This indicated a higher purity in the adult potato beetle chitin. Hence, it was concluded that a pest such as the potato beetle can be utilized as a new source of chitin [89].

In the same year, Kaya et al. studied cockchafer beetles (*Melolontha melolontha*) and compared them with crustaceans (*Oniscus asellus*) based on their chitin isolation characterization. The same procedure was followed for chitin isolations for both species. First, HCl was used to remove minerals in the organisms, and the protein structure was then removed using NaOH. Chitin obtained from these two species was characterized physicochemically. The physicochemical properties of chitins isolated from the insects and crustaceans were compared. The chitin content for the dry weights of *M. melolontha* and *O. asellus* were recorded as 13%–14% and 6%–7%, respectively. The results of FTIR, TGA, and XRD analysis were found to be more or less similar. The surface morphologies of chitins were examined via environmental scanning electron microscopy and nanofibers, and pore structures were observed. While the chitin nanofibers of *O. asellus* were adherent to each other, the nanofibers of *M. melolontha* were non-adherent. On the other hand, the number of pores was much higher in the chitin from *M. melolontha* than in the chitin from *O. asellus*. Looking at the elemental analysis results, the *M. melolontha* chitin was found to be purer than *O. asellus* chitin. For this reason, *M. melolontha* has more potential as an alternative source of chitin [90].

A method presented by Arbia et al. for fast chitin extraction from shells of crabs, crayfish, and shrimp was modified by Kaya et al. [63]. The main difference between the new method and the conventional method is the inclusion of two sodium hypochlorite (NaClO) treatments for 10 min before both the demineralization and deproteinization processes. After NaClO treatment, 15 min was enough for demineralization and 20 min was enough for the deproteinization processes. Newly extracted chitin from crab, crayfish, and shrimp shells and commercial chitin were characterized; it was observed that chitin isolated with the new method and the conventionally extracted chitin had almost the same physicochemical properties. The advantage of the new method is the relatively rapid chitin extraction. When compared to the traditional chitin extraction method, the proposed method appears to be promising regarding its time- and energy-saving nature [91].

Marei et al. used chitin isolated from four different sources: *Penaeus monodon, Schistocerca gregaria, Apis mellifera*, and *Calosoma rugosa*. Chitin was deacetylated and purified for isolating chitosan. The study revealed that the locust (*S. gregaria*) had the highest yield of chitin (12.2%), followed by shrimp (*P. monodon*), beetles (*C. rugosa*), and honeybee (*A. mellifera*), with a yield of 10%, 5%, and 2.5%, respectively. It was found that the degrees of deacetylation of chitosan obtained from locust, beetle, and honeybee cuticles were comparable (95%–98%) and higher than that of shrimp chitosan (75%). The water-binding capacity (WBC) and fat-binding capacity (FBC) of the chitosan isolated from locusts, honeybees, and beetles were comparable or lower than those of shrimp chitosan. Shrimp chitosan and honeybee chitosan had the highest ash content. The X-ray powder diffraction (XRD) showed that the chitosan isolated from locusts had the highest crystallinity. SEM analysis indicated a dense nanofiber surface structure in the shrimp, locust, and beetle chitosan and a hard, rough surface in the honeybee chitosan. For these reasons, chitin and chitosan extracted from insects are comparable to those from crustaceans [92].

Kim et al. studied the extraction of chitin and chitosan from all specimens of Type I and II two-spotted field crickets (*Gryllus bimaculatus*). The treatment was associated with the following chemical extraction with a strong acid and alkali [93]. For mineral removal, 2 N HCl was used, while 1.25 N NaOH solutions were used for deproteinization. In the deacetylation process whereby chitosan was purified, 50% NaOH (*w*/*v*) and 50% NaOH (*w*/*w*) solutions were used to further disintegrate protein associated with the main structure. The average yield of chitin and chitosan was 2.42% and 1.65% on a fresh weight basis, and 10.91% and 7.50% on a dry weight basis, respectively. The results indicate that adult exoskeletons of *G. bimaculatus* could be used as a source of chitin and chitosan. The reports suggest that chitin and chitosan are beneficial as functional additives in industrial animal feeds [93].

Ibitoye et al. yielded chitin and chitosan via chemical extraction from house crickets for amounts ranging between 4.3% and 7.1% and between 2.4% and 5.8%, respectively [94]. House crickets (*Brachytrupes portentosus*) were used without concern for the parts of the body or the sex of the insects. The chitin and chitosan obtained from the house crickets were subjected to physicochemical analysis (moisture and ash content), including elemental analysis, and were compared to commercial shrimps [94]. The authors discovered that the purified chitin and chitosan from crickets exhibit favorable similarities when compared to commercial shrimp’s chitin. It was reported that the degree of acetylation and the degree of deacetylation for both cricket and shrimp chitin and chitosan were 108.1 and 80.5%, respectively, via FTIR spectroscopy [94].

A recent study by Shin et al. was conducted on mealworm beetles and rhinoceros beetles. Chitin yields from different stages of the life cycle of the beetles (larvae, pupa, and adults) showed values between 3.9% and 14.2%. These values can be considered low compared to the crustaceans’ chitin yield. However, it was found that the chitin and chitosan from insects were nontoxic and possessed antimicrobial traits. Therefore, they are suitable as alternative material for massive production and application for commercialization [95].

Based on data collected by previous researchers (Table 1), the chitin yield content of insects in their studies favorably compared and competed with that of commercial chitin. Yields were as follows: 45% in *Bombyx eri* [85], 15%–20% in beetle larvae cuticles [10], 6.89% in *Bombus terrestris* (bumblebees) [55], 36.6% in *Periostracum Cicadae* (cicada sloughs) [86], 15% in *Holotrichia parallela* (dark black chafer beetles) [11], 26.2% in *Chyrysomya megacephala* (oriental latrine flies) [87], 20% in adult potato beetles (*Leptinotarsa decemlineata*) [89], 7% in larvae potato beetles (*Leptinotarsa decemlineata*) [89], 13%–14% in *Melolontha melolontha* (cockhafer beetles) [90], 12.2% in *Schistocerca gregaria* (desert locusts) [92], 2.5% in *Apis mellifera* (western honeybees) [92], 5% in *Calosoma rugosa* (beetles) [92], 2.35% in *Gryllus bimaculatus* (crickets) [93], 4.5%–7.1% in *Brachytrupes portentosus* (house crickets) [94], 3.9%–8.4% in mealworm beetles (*Tenebrio molitor, Zophobas morio*) [95], and 10.5%–14.2% in rhinoceros beetles (*Allomyrina dichotoma*) [95].

### 2.3. Chemical Extraction of Chitin and Chitosan from Crustaceans

Shellfish are widely cultivated around the world as important food sources, and their shells are often thrown away as industrial food waste. Hence, attention towards utilizing their waste has been increasing [97]. In this review, a total of 46 samples from crustaceans, mollusks, and other aquatic invertebrates have been collected and studied for their chemical extraction of chitin and its derivatives. The main sources of material for obtaining chitin and chitosan, especially commercially produced chitin and chitosan, are exoskeletons of crustaceans [98].

Chitin is closely associated with proteins, inorganic material that, in shellfish, consists of CaCO_3_, pigments, and lipids. Hence, to obtain chitin in its purest form, treatments for removing foreign materials associated with its structure need to be performed [7]. Table 2 is a summary of the demineralization and deproteinization processes of chitin and chitosan from crustaceans, mollusks, and other aquatic invertebrates. In this review, the 46 samples collected were taken from previous literature, wherein the majority of samples are crustaceans, such as shrimp, prawns, crabs, lobsters, and crayfish, and other types, such as cuttlefish, squid, mussels, woodlice, and barnacles.

Parameters for the chemical treatment of shellfish chitin extraction include the type of reagent, the concentration of reagent, the temperature, the solution-to-solid ratio, and the total duration of the process. However, there are other reports regarding the usage of inorganic or organic acids, such as hydrochloric acid, nitric acid, sulfuric acid, acetic acid, and formic acid. However, seeing the data collected, hydrochloric acid is the most commonly used reagent in the demineralization process. In protein treatment, reagents such as potassium hydroxide, sodium carbonate, potassium carbonate, calcium hydroxide, and sodium sulfate are applied at various concentrations in aqueous solutions. Sodium hydroxide exceeds all other reagents as the most commonly used base.

When looking at Table 2, it is clear that the most commonly used acid for removing minerals for extracting chitin from crustaceans is HCl, followed by CH_2_O_2_ and C_6_H_8_O_7_. The concentration varies from 4 to 0.25 M. While up to 100 °C of heat is applied in this step, the most common temperature used is room temperature for 10 min or even 2 days. To remove the protein associated with chitin from shrimp, prawns, crabs, lobsters, and other aquatic invertebrates, most researchers prefer to use NaOH, as Table 2 shows. The concentration range is similar to that of HCl when used for mineral removal (4–0.25 M). Demineralization is mostly performed at room temperature, and deproteinization uses a higher temperature range, from 60 to over 100 °C, for a period of 15 min or even 3 days.

The highest chitin yield can be seen from squid pen, which displays a yield of 49%. Lower amounts of minerals associated with chitin exhibited a higher chitin yield. Based on a report by Abdou et al., the highest chitin yield (cuttlefish pens) had the lowest amount of minerals. The sample was treated with 1 M NaOH for protein removal at different concentrations and for an extended period of 24 h, in addition to autoclaving, in order to increase the reaction efficiency. For treating calcium carbonate, a strong acid was used at a 1 M concentration at room temperature [99]. This was supported by Tolaimate et al., who prepared chitin and chitosan from squid and obtained a 40% chitin yield. The report showed that the mineral content of squid exhibited the lowest chitin yield [100]. The treatment involved 0.3 M acid and 0.55 M NaOH [13]. Likewise, cuttlefish pens showed the highest percentage of CaCO_3_ and the lowest chitin yield (5.4%) [99], even when the same treatment was applied.

The extraction parameters for shrimp shell chitin were optimized in 2003. A study on the kinetics for both demineralization and deproteinization, including the role of temperature in the extraction process, was also carried out to optimize the extraction methods [101]. The authors concluded that the demineralization method of chitin from shrimp shells can simply follow the variation measurement of pH in the supernatant. The increase of pH indicated higher calcium release. The raw shrimp shells were composed of approximately 20% chitin [101]. However, in their study, they claimed that excess HCl only contributes to the degradation of chitin. Therefore, they suggested that the optimal concentration of acid was 0.25 M, with a 1:40 solid-to-liquid ratio (*w*/*v*) within 15 min [101]. This was supported by another report on an optimization performed via response surface optimization, where the kinetics of chitin chemical extraction from pink shrimp (*Solenocera melantho*) shell waste were studied. The optimal deproteinization condition occurred at 75 °C, for 2.5 N NaOH. In order to have constant fluidity, a minimal solution-to-solid ratio of 5 mL/g was applied during deproteinization. Deproteinization exhibited two-stage first-order reaction kinetics. The maximum deproteinization rate constant approached 0.1 min^−1^ when the protein content was decreased from 16% to slightly above 7%. After the first 30 min, the deproteinization rate constant decreased by up to two orders of magnitude. The optimal demineralization condition was around 1.7 N HCl, with an acid solution-to-solid ratio of 9 mL/g at ambient temperature. Demineralization could be described as a pseudo-first-order reaction. The demineralization rate constant ranged from 0.00020 to 0.017 min^−1^ [56].

Compared to other crustaceans, mollusks, and other aquatic invertebrates, the sea snail chitin yield is rather average, presenting a value of 21.65%. Meanwhile, crabs, woodlice, and barnacles showed the lowest chitin yield, only ranging from 3.1% to 17% [13,90,102,103]. The strong and hard exteriors of sea snails, crabs, woodlice, and barnacles were due to the CaCO_3_ associated with the structure of chitin. Hence, due to the high amount of mineral content, the chitin yields from these samples were the lowest [13,99]. As a consequence, it is more economical to use a milder chemical, so chitin from insects has the upper hand.

### 2.4. Chemical Extraction of Chitin and Chitosan from Other Resources

As with the other samples from crustaceans, mollusks, and other aquatic invertebrates, as well as from insects, the chemical media used for isolation were a strong acid (HCl) and base (NaOH). Demineralization involved a 0.5–4 M concentration of acid at a temperature ranging from ambient to as high as 100 °C. For protein removal, the concentration of NaOH utilized was between 0.1 and 2 M for up to 3 days, at temperatures varying from room temperature to 140 °C.

The first ever reported chitin was obtained from fungi. However, since its first discovery, research development on the isolation and characterization of chitin from fungi has been scarce. Kaya et al. and Ifuku et al. have contributed additional data on the isolation of chitin from *Fomes fomentarius* (fungi) and various mushroom species (*Agaricus bisporus, Pleurotus eryngii*, *Lentinula edodes, Hypsizygus marmoreus*, and *Grifola frondosa*), as well as its characteristics [96,97]. In addition, bryozoan has structured chitin imbedded as its main constituent. Nevertheless, studies on the physiochemical properties of the chitin extracted from *Plumatella repens,* of the Bryozoa phylum, were first conducted in 2015 by Kaya et al. [96]. The chitin structure was studied comparatively by isolating chitin from an insect species (*Palomena prasina*) of the Arthropoda phylum and from *Fomes fomentarius* of the Fungi kingdom. It was observed that bryozoan chitin is in the same form as it is in the arthropod and fungal chitins. The chitin content in the dry weight of the bryozoan, fungal, and insect species was observed to be 13.3%, 2.4%, and 10.8%, respectively. The insect chitin exhibited the highest thermal stability, followed by that of the bryozoan and subsequently the fungal chitins. Surface morphologies revealed that the insect and bryozoan chitins were composed of nanofiber and pore structures, whereas the fungal chitin had no pores or fibers. The crystallinity of the insect chitin was higher than that of the bryozoan and fungal chitins. These superior characteristics of chitin from insects further support that it can compete as an alternative source of chitin in industrial aspects [96].

In addition, besides cockchafer beetles (*M. melolontha*) and potato beetles (*L. decemlineata*), Kaya et al. also produced a report on two spider species (*Geolycosa vultuosa* and *Hogna radiata*). It was claimed that, despite being a huge group of more than 44,000 species, data on the chitin structure from spiders are insufficient. In their study, it was reported that chitin isolated from these two spider species (*G. vultuosa* and *H. radiata*) represented 8%–8.5% and 6.5%–7%, respectively [116]. Environmental scanning electron microscopy (ESEM) revealed that the surface morphology of each species exhibited different traits. Chitin isolated from *G. vultuosa* has two different pore structures. One type of pore is rarely sequenced and its size ranges between 190 and 240 nm, while the second type is tightly sequenced, with pore sizes ranging from 11 to 32 nm. A new chitin surface morphology has been determined in *G. vultuosa,* and no one has reported a chitin structure with two different pore morphologies. The chitin extracted from *H. radiata* has common surface characteristics, and the nanofiber structure is in the range of 10–17 nm, with pore sizes ranging from 195 to 260 nm. The degree of acetylation values of the chitin of both *G. vultuosa* and *H. radiata* were found to be 97% and 99%, respectively [116].

## 3. Materials and Methods

A comprehensive bibliographic study was conducted to provide an in-depth insight into the purification methods of chitin, mainly involving demineralization and deproteinization. The authors carried out a literature review by means of the scientific engine Google Scholar (http://scholar.google.com), and via the databases PubMed (http://www.ncbi.nlm.gov/pubmed), Scopus (http://www.scopus.com), Sciencedirect (http://www.sciendiect.com), Elsevier (http://www.elsevier.com), MDPI (http://www.mdpi.com), Researchgate (htpps://www.researchgate.net), the American Association for Cancer Research (cancerres.aacrJ.s.org), Emeraldinsight (www.emeraldinsight.com), and Google Books (https://books.google.com). This review aims to systematically study chitin and chitosan extraction methods via chemical degradation processes previously reported around the globe to compare final yields from difference sources. The compilation of references in this review includes books, book chapters, journal articles, online articles, reports, conference proceedings, patents, documents from websites, and other web sources. References cover a time range from the 1990s to the year 2019. Since chitin was introduced in 1811, it is important to include records from these years as references. A summary of the demineralization and deproteinization processes of chitin and chitosan from other sources is tabulated to allow for a clear comparison of samples from various sources, according to their clusters.

## 4. Conclusions

The chitin and chitosan contents of insects in the studies examined in this review favorably compare and compete with those of commercial chitin and chitosan. The characteristics of chitin and its derivatives from insects are similar to those of commercial chitin from crustaceans and other aquatic invertebrates. It is also non-toxic and very safe to use. In addition, because of their large numbers and the ease of breeding, insects provide an abundant resource for larger scale chitin production. However, the mechanism and applications for chitin and chitosan from insects are still limited. To elucidate this insufficient information, further studies are needed. Therefore, according to the data reported by previous researchers, showing comparable yield values to those of crustacean chitin, and owing to the great interest in insects as an alternative source of chitin, efforts towards comprehensive knowledge in this field are of merit.

## Figures and Tables

**Figure 1 ijms-21-04978-f001:**
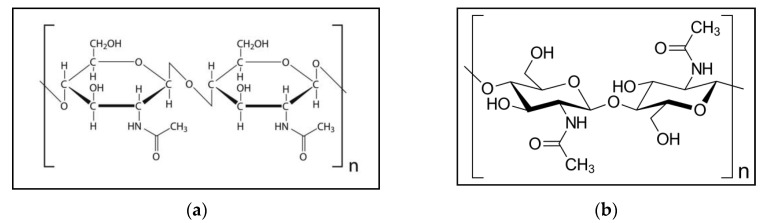
(**a**) Chemical configuration of chitin; (**b**) structure of the chitin molecule, showing two of the N-acetylglucosamine units that repeat to form long chains in the β-(1→4)-linkage.

**Figure 2 ijms-21-04978-f002:**
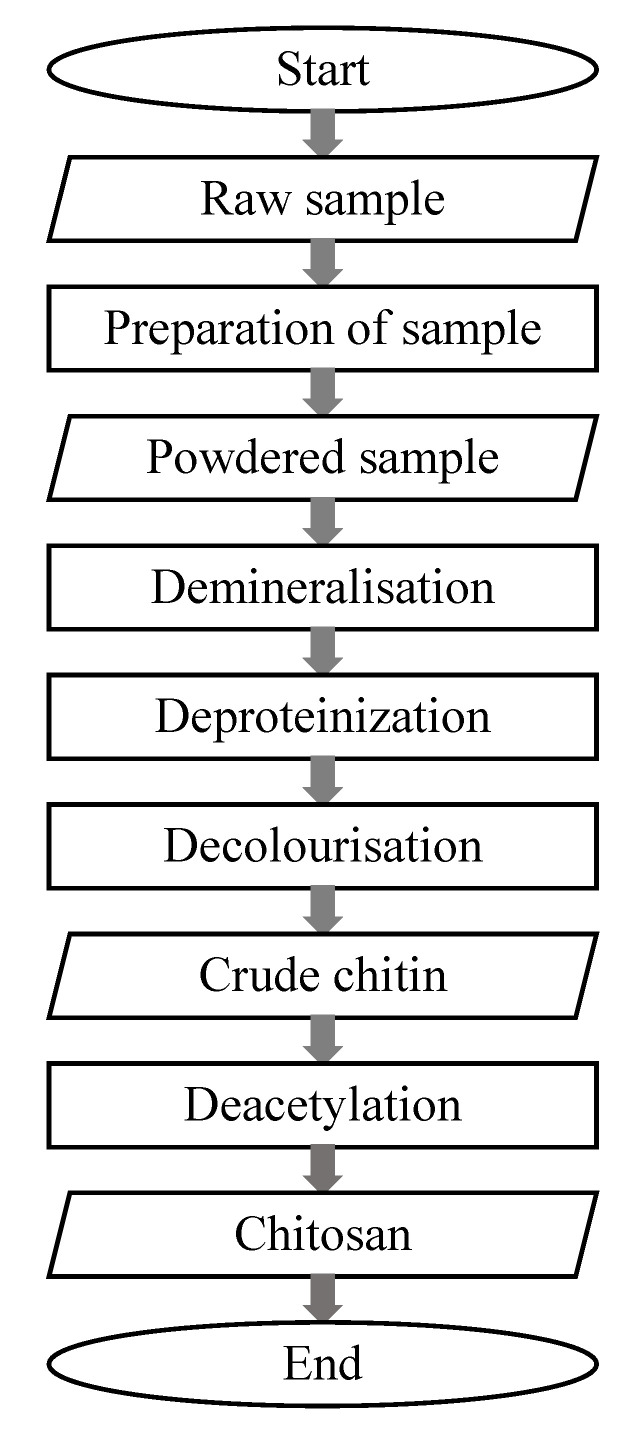
General process for chitin and chitosan extraction.

**Table 1 ijms-21-04978-t001:** A summary of the demineralization and deproteinization of chitin and chitosan from insects.

Insects														
Sources		Demineralization					Deproteinization					Yield	References	
Scientific Name	Common Name	Type of Acid	Concentration	Temperature (°C)	Solution-to-Solid Ratio (mL/g)	Duration	Type of Alkali	Concentration	Temperature (°C)	Solution-to-Solid Ratio (mL/g)	Duration			
N/A	Beetle larvae cuticle	HCl	1 N	100	N/A	20 min	NaOH	1 N	80	N/A	24–36 h	^a^ 15%–20%	[10]	
							Reflux with 0.4% of Na_2_CO_3_, 20 h							
*Bombus terrestris*	Bumblebee	HCl	1 M	100	N/A	20 min	NaOH	1 M	85	N/A	24 h	^a^ 6.89%	[55]	
*Periostracum Cicadae*	Cicada slough	HCl	1 N	100	N/A	20 min	NaOH	1 N	80	N/A	36 h	^a^ 36.6%	[86]	
							Refluxed with NaCO_3_,1 2 h							
*Holotrichia parallela*	Dark black chafer beetle	HCl	1 M	100	50	30 min	NaOH	1 M	80	50	24 h	^a^ 15%	[11]	
*Chyrysomya megacephala*	Oriental latrine fly	Oxalic acid	N/A	N/A	N/A	3 h	NaOH	1 M	95	N/A	6 h	^b^ 26.2	[87]	
*Leptinotarsa decemlineata*	Potato beetle	HCl	2 M	65–75	N/A	2 h	NaOH	2 M	80–90	N/A	16 h	^a^ 20% ^b^ 72%	[89]	
*Leptinotarsa decemlineata*	Larvae	HCl	2 M	65–75	N/A	2 h	NaOH	2 M	80–90	N/A	16 h	^a^ 7% ^b^ 67%	[89]	
*Melolontha melolontha*	Cockchafer beetle	HCl	4 M	75	N/A	2 h	NaOH	4 M	150	N/A	18 h	^a^ 13%–14%	[90]	
*Palomena prasina*	Green shield bug	HCl	2 M	100	N/A	2 h	NaOH	2 M	140	N/A	20 h	^a^ 10.8%	[96]	
*Schistocerca gregaria*	Desert locust	HCl	1 M	RT	15	N/A	NaOH	1 M	100	N/A	8 h	^a^ 12.2%	[92]	
*Apis mellifera*	Western honeybee	HCl	1 M	RT	15	N/A	NaOH	1 M	100	N/A	8 h	^a^ 2.5%	[92]	
*Calosoma rugosa*	Beetles	HCl	1 M	RT	15	N/A	NaOH	1 M	100	N/A	8 h	^a^ 5%	[92]	
*Gryllus bimaculatus*	Crickets	HCl	2 N	21	N/A	3 h	NaOH	1.25 N	95	N/A	3 h	^a^ 2.35% ^b^ 1.79%	[93]	
*Brachytrupes portentosus*	House cricket	Oxalic acid	N/A	RT	20	3 h	NaOH	1 M	95	20	6 h	^a^ 4.3%–7.1% ^b^ 2.4%–5.8%	[94]	
*Tenebrio molitor*	Mealworm beetle (larvae)	HCl	7%	25	N/A	24 h	NaOH	10%	N/A	80	24 h	^a^ 4.6% ^b^ 80%		[95]
*Tenebrio molitor*	Mealworm beetle (adult)	HCl	7%	25	N/A	24 h	NaOH	10%	N/A	80	24 h	^a^ 8.4% ^b^ 78.33%		[95]
*Zophobas morio*	Superwrm	HCl	7%	25	N/A	24 h	NaOH	10%	N/A	80	24 h	^a^ 3.9% ^b^ 83.33%		[95]
*Allomyrina dichotoma*	Rhinoceros beetle (larvae)	HCl	7%	25	N/A	24 h	NaOH	10%	N/A	80	24 h	^a^ 10.5% ^b^ 83.37%		[95]
*Allomyrina dichotoma*	Rhinoceros beetle (pupa)	HCl	7%	25	N/A	24 h	NaOH	10%	N/A	80	24 h	^a^ 12.7% ^b^ 83.37%		[95]
*Allomyrina dichotoma*	Rhinoceros beetle (adult)	HCl	7%	25	N/A	24 h	NaOH	10%	N/A	80	24 h	^a^ 14.2% ^b^ 75%		[95]
*Bombyx mori*	Silkworm	HCl	1 N	100	N/A	20 min	NaOH	1 N	80	N/A	24–36 h	^a^ 15%–20% ^b^ 70%–80%		[10]
							Reflux with 0.4% of Na_2_CO_3_, 20h							
*Bombyx mori*	Silkworm	HCl	1 M	100	10	20 min	NaOH	1 M	80	10	24 h	^a^ 2.59%–4.23% ^b^ 73%–96.75%		[88]
*Bombyx eri*	larvae	HCl	1 M	80	N/A	35	NaOH	1 M	80	NA	24 h	^a^ 45%		[85]

^a^ Chitin yield percentage; ^b^ chitosan yield percentage; NA: non-available; no data shown in reference.

**Table 2 ijms-21-04978-t002:** A summary of the demineralization and deproteinization of chitin and chitosan from crustaceans, mollusks, and other aquatic invertebrates.

Crustaceans, Mollusks, and Other Aquatic Invertebrates													
Sources		Demineralization					Deproteinization					Yield	References
Scientific Name	Common Name	Type of Acid	Concentration	Temperature (°C)	Solution-to-Solid Ratio (mL/g)	Duration	Type of Alkali	Concentration	Temperature (°C)	Solution-to-Solid Ratio (mL/g)	Duration		
*Solenocera melantho*	Pink shrimp	HCl	1.7 N	RT	9	N/A	NaOH	2.5 N	75	5	N/A	^a^ 22%	[56]
*Palaemon fabricius*	Shrimp	HCl	0.55 M	RT	N/A	1–24 h	NaOH	0.3 M	80–85	N/A	1 h	^a^ 22%	[13]
*Squilla mantis*	Squilla shrimp	HCl	0.55 M	RT	N/A	1–24 h	NaOH	0.3 M	80–85	N/A	1 h	^a^ 24%	[13]
*Parapenaeopsis stylifera*	Shrimp shells	HCl	0.25 M	RT	N/A	N/A	NaOH	1 M	RT	N/A	N/A	^a^ 20%	[101]
*Pandalus borealus*	Shrimp shells	HCl	1 M	100	N/A	20 min	NaOH	1 M	85	N/A	24 h	^a^ 6.89%	[55]
							Repeated several times during 24 h						
*Penaeus aztecus*	Brown shrimp shells	HCl	1 M	RT	N/A	N/A	NaOH	1 M	105–110	N/A	N/A	^a^ 21.53%	[99]
*Penaeus monodon*	Black tiger shrimp waste	HCOOH C_6_H_8_O_7_	0.25 M 0.25 M	N/A	30 28	30 min	NaOH	1 M	95	15	6 h	N/A	[104]
*Penaeus durarum*	Pink shrimp shells	HCl	1 N	100	N/A	20 min	NaOH	1 N	80	N/A	36 h	^a^ 23.72%	[86]
							Refluxed with NaCO_3_, 12 h						
*Parapenaeus longirostris*	Shrimp	HCl	1.23 N	RT	N/A	1 h	NaOH	3%	100	N/A	15 min	N/A	[105]
*Squilla mantis*	Squilla shrimp	HCl	1.23 N	RT	N/A	1 h	NaOH	3%	100	N/A	15 min	N/A	[105]
*Farfantepenaeus brasiliensis*	Pink shrimp	HCl	0.55 M	RT	N/A	1–24 h	NaOH	0.3 M	80–85	N/A	1 h	N/A	[106]
*Penaeus carinatus*	Shrimps	HCl	1%	RT	N/A	24 h	NaOH	2%–4%	+100	N/A	1 h	^b^ 35.49%	[24]
*Penaeus monodon*	Shrimps	HCl	1%	RT	N/A	24 h	NaOH	2%–4%	+100	N/A	1 h	^b^ 35.49%	[24]
*Lipopenaeus vannamei*	White shrimps	HCl	4%	RT	N/A	4 h	NaOH	5%	90	(w/v 1:10)	12 h	N/A	[107]
N/A	Shrimp shells	HCl	N/A	RT	N/A	15 min	NaOH	N/A	N/A	N/A	20 min	^a^ 14.8%	[91]
		NaClO treatments for 10 min each before demineralization and deproteinization											
*Penaeus japonicus*	Prawns	HCl	2 N	RT	50	48 h	NaOH	1 N	100	100	96 h	NA	[108]
*Macrobrachium rosenbergii*	Prawns shells	HCl	0.25 M	40	N/A	4 h	NaOH	0.25 M	N/A	N/A	4 h	^a^ 8.28%–5.02%	[109]
*Litopenaeus vannameil*	Prawn shells	HCl	0.5%–1%	25	4–10	24 h	NaOH	5%	60	8	2 h	^a^ 35.00% ^b^ 25.00%	[110]
*Penaeus monodon*	Giant tiger prawn	HCl	1 M	RT	15	N/A	NaOH	1 M	100	N/A	8 h	^a^ 10%	[92]
*Paralithodes camstschatecus*	Crabs	HCl	2 N	RT	50	48 h	NaOH	1 N	100	100	96 h	NA	[108]
*Grapsus marmoratus*	Marbled crab	HCl	0.55 M	RT	N/A	1–24 h	NaOH	0.3 M	80–85	N/A	1 h	^a^ 10%	[13]
*Portunus puber*	Red crab	HCl	0.55 M	RT	N/A	1–24 h	NaOH	0.3 M	80–85	N/A	1 h	^a^ 10%	[13]
*Maia squinado*	Spider crab	HCl	0.55 M	RT	N/A	1–24 h	NaOH	0.3 M	80–85	N/A	1 h	^a^ 16%	[13]
*Paralithodes camtschaticus*	Crab	HCl	7%	RT	N/A	48 h	KOH	5%	N/A	N/A	6 h	^a^ 12.1%	[111]
N/A	Rice-field crab shells	HCl	1 N	100	N/A	20 min	NaOH	1 N	80	N/A	36 h	^a^ 15.2%	[86]
							Refluxed with NaCO_3_, 12 h						
N/A	Crab shells	HCl	1 N	100	N/A	20 min	NaOH	1 N	80	N/A	36 h	^a^ 16.73%	[86]
							Refluxed with NaCO_3_, 12 h						
N/A	Crab	HCl	N/A	RT	N/A	15 min	NaOH	N/A	N/A	N/A	20 min	^a^ 13.4%	[91]
		NaClO treatments for 10 min each before demineralization and deproteinization											
*Portunus pelagicus*	Blue swimming crab	HCl	2 N	RT	50	48 h	NaOH	1 N	100	100	96 h	NA	[84]
*Potamon algeriense*	Crab	HCl	1 N	RT	15	6 h	NaOH	3%	121	10	20 min	^a^ 8.27% ^b^ 5.89%	[112]
							Autoclave at 15 psi						
*Macropipus holsatus*	Crab	HCl	4%	RT	15	1 h	NaOH	3%	65	20	2 h	^a^ 12.23% ^b^ 9.52%	[113]
*Panulirus japonicus*	Lobsters	HCl	2 N	RT	50	48 h	NaOH	1 N	100	100	96 h	NA	[108]
*Homarus vulgaris*	Lobster	HCl	0.55 M	RT	N/A	1–24 h	NaOH	0.3 M	80–85	N/A	1 h	^a^ 17%	[13]
*Scyllarus arctus*	Locust lobster	HCl	0.55 M	RT	N/A	1–24 h	NaOH	0.3 M	80–85	N/A	1 h	^a^ 25%	[13]
*Palinurus vulgaris*	Spiny lobster	HCl	0.55 M	RT	N/A	1–24 h	NaOH	0.3 M	80–85	N/A	1 h	^a^ 32%	[13]
*Astacus fluviatilis*	Crayfish	HCl	0.55 M	RT	N/A	1–24 h	NaOH	0.3 M	80–85	N/A	1 h	^a^ 36%	[13]
*Procambarus clarkii*	Crayfish	HCl	0.55 M	RT	N/A	1–24 h	NaOH	0.3 M	80–85	N/A	1 h	^a^ 20.6%	[13]
N/A	Crayfish	HCl	N/A	RT	N/A	15 min	NaOH	N/A	N/A	N/A	20 min	^a^ 15.3%	[91]
		NaClO treatments for 10 min each before demineralization and deproteinization											
*Sepia officinalis*	Cuttlefish	HCl	0.55 M	RT	N/A	1–24 h	NaOH	0.3 M	80–85	N/A	1 h	^a^ 20%	[13]
*Loligo vulgaris*	Squid	HCl	0.55 M	RT	N/A	1–24 h	NaOH	0.3 M	80–85	N/A	1 h	^a^ 40%	[13]
N/A	Cuttlefish pens	HCl	1 M	RT	N/A	N/A	NaOH	1 M	105–110	N/A	N/A	^a^ 5.4%	[99]
N/A	Squid pens	HCl	1 M	RT	N/A	N/A	NaOH	1 M	105–110	N/A	N/A	^a^ 49%	[99]
N/A	Mussel shells	HCl	0.68 M	RT	10	6 h	NaOH	0.62 M	N/A	10	16 h	^a^ 23.25%	[114]
*Oniscus asellus*	Woodlouse	HCl	4 M	75	N/A	2 h	NaOH	4 M	150	N/A	18 h	^a^ 6%–7%	[90]
		Reflux in magnetic stirrer					Heat in magnetic stirrer						
*Lepas anatifera*	Barnacles	HCl	0.55 M	RT	N/A	1–24 h	NaOH	0.3 M	80–85	N/A	1 h	^a^ 7%	[13]
*Chelonibia patula*	Barnacles	HCl	1 M	N/A	N/A	10 min	NaOH	2 M	N/A	N/A	20 min	^a^ 3.1%	[102]
*Daphnia magna*	Water flea	HCl	1 M	65–75	N/A	6 h	NaOH	1 M	65	N/A	20h	^a^ 13%–21%	[115]
*Conus inscriptus*	Sea snail	HCl	1 M	60	N/A	30 min	NaOH	3 M	80	N/A	2 h	^a^ 21.65%	[103]

^a^ Chitin yield percentage; ^b^ chitosan yield percentage; NA: non-available; no data shown in reference.

**Table 3 ijms-21-04978-t003:** A summary of the demineralization and deproteinization of chitin and chitosan from other resources.

Others													
Sources		Demineralization					Deproteinization					Yield	References
Scientific Name	Common Name	Type of Acid	Concentration	Temperature (°C)	Solution-to-Solid Ratio (mL/g)	Duration	Type of Alkali	Concentration	Temperature (°C)	Solution-to-Solid Ratio (mL/g)	Duration		
*Pleurotus eryngii*	Mushrooms	HCl	2 M	RT	N/A	48 h	NaOH	2%	100	N/A	48 h	N/A	[117]
*Agaricus bisporus*	Mushrooms	HCl	2 M	RT	N/A	48 h	NaOH	2%	100	N/A	48 h	N/A	[117]
*Lentinula edodes*	Mushrooms	HCl	2 M	RT	N/A	48 h	NaOH	2%	100	N/A	48 h	N/A	[117]
*Grifola frondosa*	Mushrooms	HCl	2 M	RT	N/A	48 h	NaOH	2%	100	N/A	48 h	N/A	[117]
*Hypsizygus marmoreus*	Mushrooms	HCl	2 M	RT	N/A	48 h	NaOH	2%	100	N/A	48 h	N/A	[117]
N/A	Chicken feet	HCl	0.5–1.5 N	N/A	10–20	N/A	NaOH	0.1–2 N	N/A	5–20	N/A	^a^ 15.2%	[70]
*Geolycosa vultuosa*	Spider	HCl	4 M	60–65	N/A	1 h	NaOH	1 M	130–135	N/A	16 h	^a^ 8%–8.5%	[116]
*Hogna radiata*	Spider	HCl	4 M	60–65	N/A	1 h	NaOH	1 M	130–135	N/A	16 h	^a^ 6.5%–7%	[116]
*Labeo rohit*	Fish scales	HCl	1%	RT	N/A	36 h	NaOH	2 N	RT	N/A	36 h	N/A	[118]
*Labeo rohit*	Fish scales	HCl	1%	RT	N/A	N/A	NaOH	0.5 N	RT	N/A	18 h	N/A	[119]
*Plumatella repens*	Creeping bryozoan or moss animal	HCl	2 M	100	N/A	2 h	NaOH	2 M	140	N/A	20h	^a^ 13.3%	[96]
*Fomes fomentarius*	Fungi	HCl	2 M	100	N/A	2 h	NaOH	2 M	140	N/A	20h	^a^ 2.4%	[96]
*Zophobas morio*	Superworm larvae	HCl	1 M	35	20	30 min	NaOH	0.5–2 M	80	20	20h	^a^ 4.77%–5.43% ^b^ 65.84%–75.52%	[120]

^a^ Chitin yield percentage; ^b^ chitosan yield percentage; NA: non-available; no data shown in reference.

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
