# Peer review of "The Potential of Insects as Alternative Sources of Chitin: An Overview on the Chemical Method of Extraction from Various Sources"

_ijms, 2020, doi:10.3390/ijms21144978_

Round 1
Reviewer 1 Report
The review of Abidin and coworkers summarize chemical extraction procedures for the isolation of chitin from insects. Furthermore, the review provide information about the structural differences between chitin, chitosane and cellulose and discuss the consequences for the isolation. The topic of the manuscript is interesting, but the manuscript needs to be rewritten carefully. Unfortunately, the manuscript includes a lot of redundancy and also the general structure should be improved. Please find enclosed some general comments, which needs to be addressed.
Abstract:
In my opinion, you should shorten the abstract.
Line 23 and 31 (just two examples): Please rephrase “Based on previous data by previous researchers”
Which criteria have you used for your bibliographic research? Which years does the literature search cover?
Introduction:
The introduction needs to be restructured and includes a lot of redundancy.
Line 82: Remove “resource”
Line 132: What do you mean with structurally similar?
Results:
I think for a review you don´t need a result section.
Discussion:
It would be interesting to know what happens to the secondary structure during the extraction. Does it remain with some methods? There are several methods described for this purpose at least for sponges.
Klinger et al. 2019 “It would be interesting to know what happens to the secondary structure during ther extraction. Does it remain with some methods?”
Shaala et al. 2019 “New source of 3D chitin scaffolds: the Red Sea demosponge Pseudoceratina arabica (Pseudoceratinidae, Verongiida)”
Are there also microwave based extraction procedures available for insects? You should at least discuss this.
What’s the difference between table 1 and 2? The same literature is included and information are overlapping.
You should also focus on new publications. Just as examples:
Machalowski et al. 2019 “Spider Chitin: An Ultrafast Microwave-Assisted Method for Chitin Isolation from Caribena versicolor Spider Molt Cuticle”
General comment:
Check notation of chemical formulas in the text: CaCO3 --> CaCO3
Author Response
Response to Reviewer 1 Comments
The review of Abidin and coworkers summarize chemical extraction procedures for the isolation of chitin from insects. Furthermore, the review provide information about the structural differences between chitin, chitosane and cellulose and discuss the consequences for the isolation. The topic of the manuscript is interesting, but the manuscript needs to be rewritten carefully. Unfortunately, the manuscript includes a lot of redundancy and also the general structure should be improved. Please find enclosed some general comments, which needs to be addressed.
ABSTRACT
Point 1:
In my opinion, you should shorten the abstract. Line 23 and 31 (just two examples):
Response 1: The abstract is cut short. For Line 23-31, the examples given are left to two examples only.
Point 2: Please rephrase “Based on previous data by previous researchers”
Response 2: Rephrased to ‘based on previous data’
Point 3: Which criteria have you used for your bibliographic research? Which years does the literature search cover?
Response 1: method section is added (L137-153). All information is added there.
INTRODUCTION
Point 4: The introduction needs to be restructured and includes a lot of redundancy.
Response 4: Introduction has been restructured and redundancy was removed.
Point 5: Line 82: Remove “resource”
Response 5: ‘resource’ has been removed
Point 6: Line 132: What do you mean with structurally similar?
Response 6: chemical structure for both chitosan and cellulose.
RESULTS
Point 7: I think for a review you don´t need a result section.
Response 7: Result section was removed and replaced with methodology.
DISCUSSION
Point 8: It would be interesting to know what happens to the secondary structure during the extraction. Does it remain with some methods? There are several methods described for this purpose at least for sponges.
Klinger et al. 2019 “It would be interesting to know what happens to the secondary structure during ther extraction. Does it remain with some methods?”
Shaala et al. 2019 “New source of 3D chitin scaffolds: the Red Sea demosponge Pseudoceratina arabica (Pseudoceratinidae, Verongiida)”
Response 8: Changes has been made.
Point 9: Are there also microwave based extraction procedures available for insects? You should at least discuss this.
Response 9: Changes has been made.
Point 10: What’s the difference between table 1 and 2? The same literature is included and information are overlapping.
Response 10: Table 1 has been removed.
Point 11: You should also focus on new publications. Just as examples:
Machalowski et al. 2019 “Spider Chitin: An Ultrafast Microwave-Assisted Method for Chitin Isolation from Caribena versicolor Spider Molt Cuticle”
Response 11: Recent citations has been added.
GENERAL COMMENT
Point 12: Check notation of chemical formulas in the text: CaCO3 --> CaCO3
Response 12: Changes has been made
Reviewer 2 Report
The article is potentially written deciphering the “Potential of insects as alternative sources of chitin: An overview on the chemical method of extraction from various sources” by Abidin et al. However, the manuscript has to be improved to make it more suitable for acceptance in the journal. The following questions need to be addressed.
Comment 1: In abstract, L25-31, rewrite the sentence.
Comment 2: L23-25 and L31-L33 are redundant sentences. Please rewrite. the abstract, they tried to include all results, but it looks not good.
Comment 3: In introduction, L40-42, please include some recent references.
Comment 4: Instead of L55, give reference for L60.
Comment 5: Results and Materials and methods sections are too small. Make it in detail or remove.
Comment 6: L65, L296 give line space for paragraph.
Comment 7: L212 to L219, rewrite the paragraph.
Comment 8: Please check chemical formulas throughout the manuscript. For example: CH3-OH should be CH3-OH; CaCO3 should be CaCO3.
Comment 9: Figure 3 is not catchable, please make new one.
Comment 10: In discussion, L153-223, there is no section. Suddenly from L226 3.1. Why? Otherwise give section for previous paragraphs.
Comment 11: Section 3.1, please include some recent references about commercial preparation of chitin.
Comment 12: Please check citation in author instructions. Suddenly found author name instead of number. For example: Zhang et al., (2000); Matjan et al., (2007); Song et al., (2013). Please check throughout the manuscript.
Comment 13: Please check numerical writing throughout the manuscript. For example, in L257 you mentioned 1 N HCl but in L267 2N, here so space. Same way for hours, minutes,︒C and %. Please read author instructions carefully. Check tables also.
Comment 14: L380, please rewrite the sentence. You should not use “His group….”.
Comment 15: L388-89, please rewrite the sentence.
Comment 16: L460 and 464, please change min-1 to min-1. Check throughout the manuscript.
Comment 17: L482-483; L487; L497, L498, L501 and throughout the manuscript, Scientific name should be in Italics. Please change.
Comment 18: Please reduced number of sentences in discussion too much of information with lack of novelty.
Comment 19: Please rewrite materials and methods or remove.
Comment 20: Please rewrite the conclusion. Four lines not enough. Please check English throughout the manuscript.
Author Response
Response to Reviewer 2 Comments
The article is potentially written deciphering the “Potential of insects as alternative sources of chitin: An overview on the chemical method of extraction from various sources” by Abidin et al. However, the manuscript has to be improved to make it more suitable for acceptance in the journal. The following questions need to be addressed.
Comment 1: In abstract, L25-31, rewrite the sentence.
Response 1: Revised. The examples are left to only two examples.
Comment 2: L23-25 and L31-L33 are redundant sentences. Please rewrite. the abstract, they tried to include all results, but it looks not good.
Response 2: Redundant sentences are removed. The examples are left to only two examples.
Comment 3: In introduction, L40-42, please include some recent references.
Response 3: Recent references has been added
Comment 4: Instead of L55, give reference for L60.
Response 4: Changes has been made.
Comment 5: Results and Materials and methods sections are too small. Make it in detail or remove.
Response 5: Results section was removed and replaced with methods section.
Comment 6: L65, L296 give line space for paragraph.
Response 6: Changes has been made.
Comment 7: L212 to L219, rewrite the paragraph.
Response 7: The paragraph has been rewritten.
Comment 8: Please check chemical formulas throughout the manuscript. For example: CH3-OH should be CH3-OH; CaCO3 should be CaCO3.
Response 8: Changes has been made.
Comment 9: Figure 3 is not catchable, please make new one.
Response 9: Remade.
Comment 10: In discussion, L153-223, there is no section. Suddenly from L226 3.1. Why? Otherwise give section for previous paragraphs.
Response 10: Because it is the general discussion. As in the introduction to discussion if that makes sense.
Comment 11: Section 3.1, please include some recent references about commercial preparation of chitin.
Response 11: Recent citation added.
Comment 12: Please check citation in author instructions. Suddenly found author name instead of number. For example: Zhang et al., (2000); Matjan et al., (2007); Song et al., (2013). Please check throughout the manuscript.
Response 12: Changes has been made.
Comment 13: Please check numerical writing throughout the manuscript. For example, in L257 you mentioned 1 N HCl but in L267 2N, here so space. Same way for hours, minutes,︒C and %. Please read author instructions carefully. Check tables also.
Response 13: Changes has been made.
Comment 14: L380, please rewrite the sentence. You should not use “His group….”.
Response 14: Changes has been made.
Comment 15: L388-89, please rewrite the sentence.
Response 15: Sentence has been rewritten
Comment 16: L460 and 464, please change min-1 to min-1. Check throughout the manuscript.
Response 16: Changes has been made.
Comment 17: L482-483; L487; L497, L498, L501 and throughout the manuscript, Scientific name should be in Italics. Please change.
Response 17: Changes has been made.
Comment 18: Please reduced number of sentences in discussion too much of information with lack of novelty.
Response 18: Changes has been made.
Comment 19: Please rewrite materials and methods or remove.
Response 19: Rewritten.
Comment 20: Please rewrite the conclusion. Four lines not enough. Please check English throughout the manuscript.
Response 20: Conclusion has been rewritten. English checking is in the process by MDPI English Editing Services.
Round 2
Reviewer 2 Report
Comment 1: Still, did not find recent references in the Introduction section and Discussion section. References are from the 1970s and 80s and 2012-14. Please include recent references.
Comment 2: Not satisfied with Figure 3. Please make a good one.
Comment 3: Change discussion as "Results and discussion".
Comment 4: Please rewrite the conclusion. Please check the English.
Comment 5: Please read author instructions carefully and write numbers and so on.
